# Financial Analysis of the Use of Land: Agriculture or Woodlot

Jacqueline Ninson *[ID], Irene S. Egyir, Akwasi Mensah-Bonsu and Edward Ebo Onumah [ID]

Department of Agriculture Economics and Agribusiness, College of Basic and Applied Science,
University of Ghana, Legon-Accra P.O. Box LG 25, Ghana; iegyir@ug.edu.gh (I.S.E.);
AMensah-Bonsu@ug.edu.gh (A.M.-B.); eeonumah@ug.edu.gh (E.E.O.)
* Correspondence: jbaidoo013@st.ug.edu.gh; Tel.: +233-501380932

**Abstract:** Agriculture is the main driver of deforestation. In other to reduce deforestation, a viable alternative livelihood strategy, aside from agriculture, must be in place to provide a sustainable income for investors. Managing forests for sustainable production (the forest economy) has been suggested as an alternative for sustainable land use practice. In the current study, we undertook a comparative analysis of woodlots and agriculture. The profitability of agriculture and woodlot production in Ghana was compared using a profitability model. We looked at profitability in terms of Net Present Value (NPV) and the Benefit-Cost Ratio (BCR) of three regions in Ghana, namely, Ashanti, Bono-East, and Western Regions. We found that woodlot producers with contractual relationships with the Forest Commission and other forestry companies produce the highest Net Present Value (NPV) and Benefit-Cost Ratio (BCR). However, this profitability is marginally higher than that of agriculture, which gives a fixed yearly return. This means woodlot production may not be a panacea to reducing agriculture in Ghana.

**Keywords:** agriculture; woodlot; profitability; deforestation; land use

## 1. Introduction

Deforestation is occurring globally, with half of the illegal deforestation caused by clearing forest land for agriculture [1]. This situation is worsening in the 21st century. From 2000 to 2010, the rate of deforestation in Ghana was very high (2%) compared to that of other West African countries (0.6%) [2]. Since 1987, about 70% of the deforestation in Ghana can be attributed to agriculture expansion [3].

In Ghana, forest management and conservation are vested by the government and often managed by the community [4]. However, deforestation has occurred both on-reserve and off-reserve. This may be because, unlike agriculture, the forest does not provide sustainable revenue for investors [5]. People use forestland for agriculture expansion since they do not know the value of managing forests (i.e., woodlot production) [6]. This leads to deforestation as forestland is used for agriculture [2,3]. The forest is depleted because people who have access to the forest do not receive an economic benefit from the forest [6].

Land is a primary resource or input in most production processes, such as woodlots and expansion of agriculture [7], but it is also a limited and fixed resource. According to production theory [8], an investor is faced with the choice of more than one enterprise to invest in. Holding all other factors constant, a rational producer is expected to choose the enterprise that maximizes their profit, whether that is, for example, a woodlot or clearing for agriculture [9]. In this assumption, production theory indirectly requires that the producer has enough information about the available alternatives to make an informed decision. In recent times, forestlands have been used for agriculture [10]. This may be because agriculture is more profitable than woodlots or the producers are not fully aware of the profitability of woodlots.

Timber is the most economic product in the forest [11]. In order to reduce deforestation, ref. [6] suggests that forests should be managed (forests planted and timber logs harvested

when the tree is mature) in such a manner that revenue is generated sustainably from timber. Investors are, however, indecisive about investing in the woodlots [12,13]. The financial cost and benefit of turning forestlands into farmlands or woodlot production is unclear; that is, it is not clear whether one should invest in woodlots or use forestland for agriculture expansion. This study aimed to estimate the profitability of managed forest and agriculture expansion to farmers in Ghana.

## 2. Materials and Methods

### 2.1. Research Design and Sampling Methodology

This study used a comparative research design to compare the profitability of woodlots and agricultural production as the main land uses. The study used a multi-stage sampling methodology. In the first stage, one area with a forest reserve was randomly selected from each agro-ecological zone [14]. The forest reserves were considered because deforestation is driven by agricultural expansion in the forest reserves [15]. Three regions, namely, Ashanti, Bono-East, and Western Regions, were selected (Table 1). This was because these three regions, in addition to having a forest reserve, had the highest rates of deforestation between 2001 and 2020 [16]. In the second stage, districts in the regions were purposively selected from each of the three regions. In Western Region, Wassa East was selected because it has the largest and most diverse forest reserve (Subri forest reserve) experiencing deforestation [17] in Ghana. In Wassa East and Tarkwa Nsuaem, farmers undertake taungya farming. Taungya farming is a collaboration between farmers and the Forest Commission where the land on the reserve is given to farmers to farm on for a maximum of three years (after which trees form canopies, thus preventing farming). Farmers intend to plant timber trees while farming on the land. Atebubu was selected because it has one of the largest associations of woodlot producers. Mampong was selected because it is the location of a forest company that has been given a consignment to plant trees.

**Table 1.** Study areas.

| Agro-Ecological Zone | Region | District | Communities |
|---|---|---|---|
| Forest | Ashanti Region | Mampong<br>Sekyere Afram Plains | Mampong<br>Drobonso |
| Transitional | Bono-East Region | Kintampo South<br>Atebubu | Adamsanu, Adum Anafoɔ<br>Beposo, Kwame Danso |
| Coastal | Western Region | Wassa East | Sekyere Krobo, Kakoase, Essamang,<br>Nsuta, Sekyere Aboaboso |
| | | Tarkwa Nsuaem | Agona, Kakoase |

For communities, a change detection yielded a hotspot of deforestation in the various districts selected. Communities were selected based on nearness to reserves and biodiversity. After the selection of hotspots, stakeholders ranked the communities. The Forest Commission and the Ministry of Food and Agriculture provided a list of investors in the various communities. Respondents were randomly selected from the list of investors given.

### 2.2. Study Area

This study was conducted in six districts, namely, Kintampo South, Atebubu Amantin, Mampong, Sekyere Afram Plains, Wassa East and Tarkwa districts (Figure 1).

Mampong: Mampong has a population of 88,051, of which 61% are rural. About 54% of the people are employed in the agricultural, forestry, and fishery sectors. About 96% of the people who are employed in the agriculture sector cultivate crops [18].

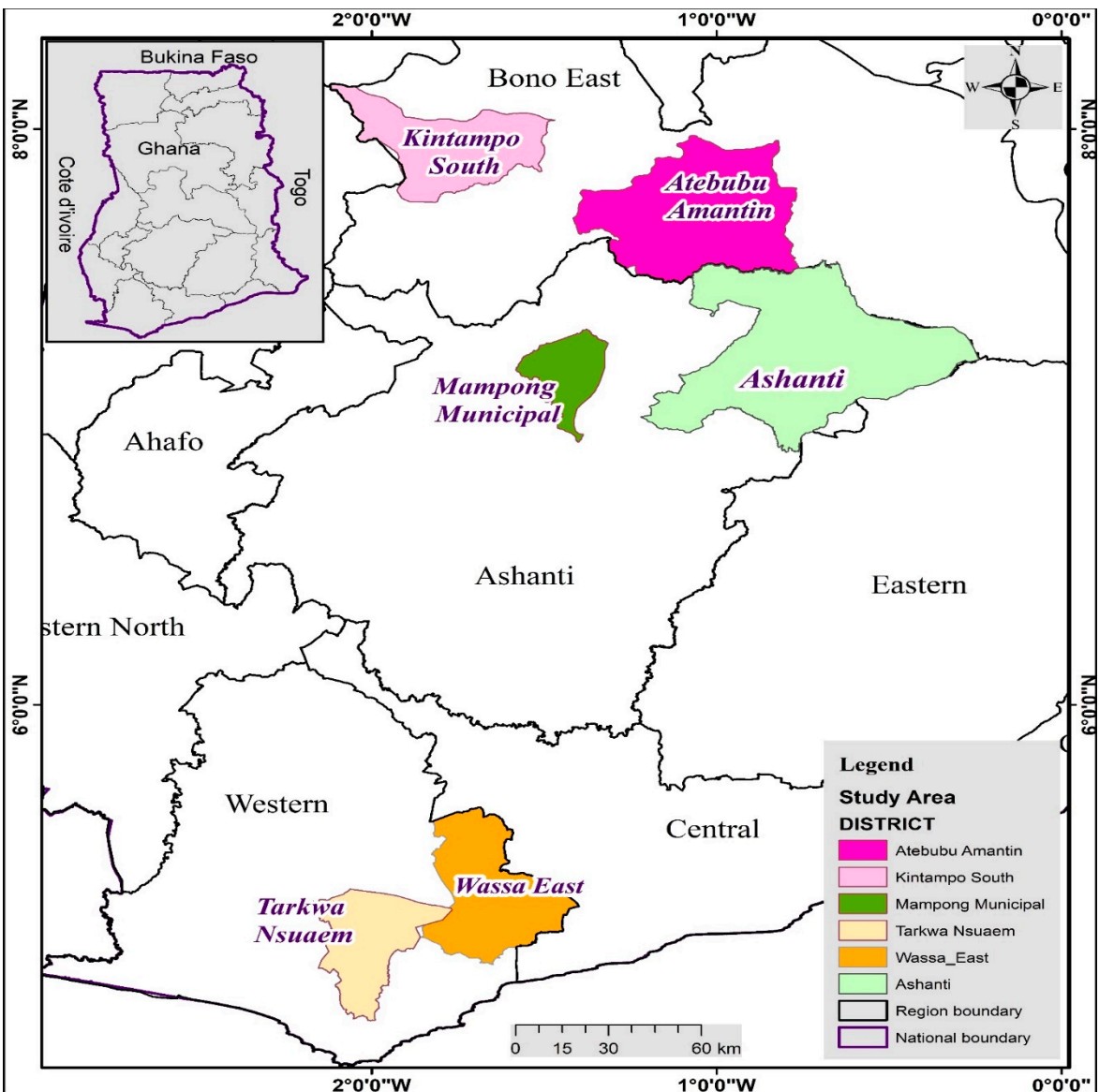

**Figure 1.** Study Area.

Sekyere Afram Plains: The district covers an estimated land area of 3525.1 square kilometers, representing 14.5 per cent of the regional land size of 24,389 square kilometers. Sekyere Afram Plains has a population of 28,535 comprising 5411 households. About 89.6 per cent of those employed are engaged as skilled agricultural, forestry, and fishery workers. As many as 81.4 per cent of households in the district are engaged in agriculture, which is mostly in the form of crop production [19].

Kintampo south: About 90 per cent (88.3%) of households in the district are engaged in agriculture, mainly crop production. Agriculture is the major economic activity in the Kintampo South District in terms of employment and income generation [20].

Atebubu-Amanten District: Atebubu has a population of 105,938, of which 53% are rural. There are 20,349 households with an average household size of 5.1. About 66% of employed people are in agriculture, forestry, and fishery sectors. As many as 70.2 per cent of households in the district are engaged in agriculture, and more than 95% of people engaged in agriculture are involved in crop production [21].

Wassa East: The population of Wassa East is 81,073, and the district has a total land area of 1651.992 square kilometers. Agriculture is the major occupation of the inhabitants of the district. The major staple food crops produced in the district include cassava, plantain,

maize, cocoyam, and vegetables. The predominant cash crops are cocoa, oil palm, and rubber. Crop farming (96.1%) is the major activity undertaken by households engaged in agriculture [22].

Tarkwa Nsueam: Tarkwa has a population of 90,477, of which 31.5 per cent are engaged as skilled agricultural, forestry, or fishery workers. The municipality has a total land area of 905.2 square km and has Tarkwa as its capital. Tarkwa Nsuaem has large forest reserves, such as the Bonsa Reserve, Ekumfi Reserve, Neung South Reserve, and Neung North Reserve [23].

### 2.3. Demographics of Respondents

Males are dominant in both farming activities and woodlot production, as more than half (50%) of males are involved in farming and woodlot production (Table 2). This may be because both agriculture and woodlot production require strength, which therefore attracts men.

**Table 2.** Demographic and socio-economic characteristics of respondents.

| Characteristic | Farmers | | Woodlot Producers | | t-Statistic |
| --- | --- | --- | --- | --- | --- |
| | Mean or Percentages | Std. Dev | Mean or Percentages | Std. Dev | |
| Respondent is Male (yes or no) | 110 (57.1) | 0.54 | 116 (60) | 0.48 | 0.2 |
| Age of the respondent in years | 44.3 | 13.46 | 51 | 12.11 | 0.08 |
| Respondent is married (yes or no) | 152 (78.6) | 0.52 | 160 (82.9) | 0.57 | 0.66 |
| Respondent is a Christian | 179 (92.9) | 0.59 | 182 (94.3) | 0.45 | 0.19 |
| Sex of household head (male) | 151 (76.3) | 0.04 | 149 (74.3) | 0.04 | 0.55 |
| Years of education of the respondent | 7.1 | 0.40 | 6.1 | 0.71 | 0.91 |
| Number of people in the household | 4.9 | 2.98 | 5.5 | 3.27 | 0.08 |
| Household head is female | 48 (25) | 0.42 | 28 (14.3) | 0.55 | 0.36 |
| Other source of income aside from agriculture (yes) | 116 (52) | 0.04 | 106 (48) | 0.04 | 0.02 |
| Land sizes owned | 3.20 | 0.06 | 7.63 | 0.05 | 0.04 |
| Participation on forum | 158 (56) | 0.04 | 124 (44) | 0.04 | 0.86 |

Woodlot and agriculture farmers are mostly comparable, with the exceptions of age and household size. The average age of farmers is 44 years, whereas that of woodlot producers is 51 years (Table 2). More than three-fourths of respondents in both agriculture and woodlot production are married (Table 2). This may explain why most (more than three-fourths) of the household heads are males (Table 2). Almost all (more than 90%) of the respondents are Christians (Table 2). The household size of woodlot producers is slightly higher than that of farmers (5.5 compared to 4.9, respectively) (Table 2). This may explain why farmers enter into the MTS scheme, i.e., to provide enough food for their families. The years of education of farmers are slightly higher than those of woodlot producers (7.1 compared to 6.1, respectively) (Table 2). Investors with a large amount of land (more than 7 hectares) prefer woodlot production to agriculture (Table 2).

### 2.4. Data Collection Tools

Cross-sectional primary data was collected using a household/individual questionnaire from 386 respondents. The sample consisted of 193 woodlot producers and 193 agricultural producers. Data were collected between September 2021 and January 2022.

### 2.5. Data Analysis

This study used profitability analysis to separately determine the profitability [24] of woodlot and agricultural land use. The components of profitability are the costs and the revenues for each enterprise within a defined timeframe (Equation (1)).

$$NPV = \sum\nolimits_{t=0}^{n} \frac{(c_t - b_t)}{(1+k)^t} \tag{1}$$

where $C_t$ represents the cash inflows in each project year and $b_t$ represents the cash outflows in each project year; $n$ is the number of years; and $k$ is the interest (discount) rate [25].

The selection criterion is to accept all independent projects with *NPV* of zero or greater, at a specified discount rate. A negative *NPV* implies that, at the assumed opportunity cost of capital, the present worth of the benefit stream is less than the present worth of the cost stream, meaning the enterprise will be unable to recover its investments [24]. The net present value is calculated as in Equation (2).

$$BCR = \sum_{t=0}^{n} \frac{(c_t/b_t)}{(1+k)^t} \tag{2}$$

where $C_t$ represents the cash inflows in each project year and $b_t$ represents the cash outflows in each project year; $n$ is the number of years; and $k$ is the interest (discount) rate [25].

The selection criterion is to accept all independent projects with a *BCR* of 1 or greater, at a specified discount rate. A Benefit-Cost Ratio (BCR) below 1 implies that, at the assumed opportunity cost of capital, the present worth of the benefit stream is less than the present worth of the cost stream, meaning the enterprise will be unable to recover its investments

## 3. Results

The dominant staple crops grown in the Ashanti, Bono-East, and Western Regions are maize, yam, and plantain, respectively (Tables A1–A3). In terms of cash crops, cocoa is grown in Ashanti and Western Regions, and cashew is grown in the Bono-East Region (Tables A1–A3). These findings are evident in [18–23]. The Ashanti Region has the highest sales in both staple and cash crops (GHS[1] 18,350.8 and GHS 19,500) compared to Bono-East and Western Regions (Table 3). This may be because the Ashanti Region is located in a forest zone dominated by a good climate and soil [14].

**Table 3.** Profit and loss statement for agricultural investors in the Ashanti, Bono-East, and Western Regions of Ghana.

| | Ashanti Region | | Bono-East Region | | Western Region | | Average | |
|---|---|---|---|---|---|---|---|---|
| | Staple Crop | Cash Crop | Staple Crop | Cash Crop | Staple Crop | Cash Crop | Staple Crop | Cash Crop |
| | GHS | GHS | GHS | GHS | GHS | GHS | GHS | GHS |
| Sales | 18,350.8 | 19,500 | 14,781 | 3569.76 | 6430 | 5610 | 13,187.25 | 9559.919 |
| Less Cost of Sales | 1821.45 | 1600 | 1605.67 | 215.786 | 1420 | 400 | 1615.705 | 738.5952 |
| Gross profit | 16,529.3 | 17,900 | 13,175.3 | 3353.972 | 5010 | 5210 | 11,571.54 | 8821.324 |
| Less Expenses | | | | | | | | |
| Transportation | 1319.46 | 560 | 1162.46 | 157 | 330 | 280 | 937.3033 | 332.3333 |
| Weedicides | 2283.06 | 3000 | 1884.81 | 398.25 | 900 | 1000 | 1689.293 | 1466.083 |
| Labor | 2246.65 | | 1568.08 | 678.571 | 676 | 900 | 1496.912 | 789.2857 |
| Depreciation | 158.848 | 1890 | 158.848 | 0 | 159 | 289 | 158.8987 | 726.3333 |
| Others | 1145.15 | 769 | 929.368 | 215.786 | 330 | | 801.5069 | 492.3929 |
| Net Profit | 9376.13 | 11,681 | 7471.76 | 1904.36 | 2615 | 2741 | 6487.629 | 5442.122 |
| Cash Inflow | 18,350.8 | 19,500 | 14,781 | 3569.76 | 6430 | 5610 | 13,187.25 | 9559.919 |
| Cash Outflow | 8815.78 | 5929 | 7150.38 | 1665.39 | 3656 | 2580 | 6540.72 | 3391.464 |

Taungya farming is undertaken in all three regions (Table 4) due to the availability of a forest reserve in each region.

However, the Ashanti Region is the location of a forestry company that has contractual relationships with woodlot investors. The Bono-East Region is also the location of woodlot investors who are mostly involved in teak farming and have no contractual relationship. Moreover, some woodlot investors have formed associations, whose membership is around 200. Taungya farmers have the same profit in all of the regions due to a structure designed by the Forestry Commission, in which input is provided directly by the Commission. Moreover, the Commission is directly in charge of the sales of woodlots during harvesting [26]. Since the representative of the government (i.e., the Forestry Commission) is in charge of the

sales of woodlots in the taungya system, high sales are reported. Although Table 4 shows taungya farmers have the highest profit (GHS 13,120) after contracted woodlot investors (GHS 30,865.5), there is some mistrust among taungya farmers about the disbursement of money after the sales of woodlots [26].

**Table 4.** Profit and loss statement for woodlot investors in the Ashanti, Bono-East, and Western Regions of Ghana.

| | Ashanti | | | Bono-East | | Western Region | |
|---|---|---|---|---|---|---|---|
| | On Reserve (Taungya Farmers) | Contracted Woodlot Farmers | Off Reserve | On Reserve (Taungya Farmers) | Farmers Who Have Formed Associations | On Reserve (Taungya Farmers) | Average for on Reserve (Taungya Farmers) |
| | GHS | GHS | GHS | GHS | GHS | GHS | GHS |
| Sales | 13,200 | 31,915.5 | 2000 | 13,200 | 12,000 | 13,200 | 13,200 |
| Less cost of sales | 0 | 125 | 0 | 0 | 100 | 0 | 0 |
| Gross profit | 13,200 | 31,790.5 | 2000 | 13,200 | 11,900 | 13,200 | 13,200 |
| Less Expenses | | | | | | | |
| Transportation | 0 | 100 | 0 | 0 | 0 | 0 | 0 |
| Weedicides | 0 | 250 | 0 | 0 | 300 | 0 | 0 |
| Labor | 0 | 250 | 150 | 0 | 1000 | 0 | 0 |
| Depreciation | 80 | 225 | 80 | 80 | 52.7 | 80 | 80 |
| Others | 0 | 100 | 0 | 0 | 0 | 0 | 0 |
| Net Profit | 13,120 | 30,865.5 | 1770 | 13,120 | 10,547.3 | 13,120 | 13,120 |
| Cash Inflow | 13,200 | 31,915.5 | 2000 | 13,200 | 12,000 | 13,200 | 13,200 |
| Cash Outflow | 0 | 1050 | 150 | 0 | 1400 | 0 | 0 |

### 3.1. Estimating the Profitability of Woodlot and Agriculture to Investors in Ghana

The cash outflow in cash crop production is slightly lower (GHS 3146 less) than that in staple crops (Table 5). This may be because different crops, such as plantain, cassava, and maize, are planted on the same hectare of land. Moreover, because staple crops are mostly annual crops, one pays more in a year compared to cash crops. The expenses in staple crop production are GHS 1277.486 more than those in cash crop production (Table 5). The cost of production for staple crop production is GHS 87,711 higher than that for cash crop production (Table 5).

**Table 5.** Average profit and loss statement for the year ended 2021 for agriculture and woodlot investors.

| | Staple Crop | Cash Crop | Off Reserve | On Reserve (Taungya Farmers) | Contracted Farmers |
|---|---|---|---|---|---|
| | GHS | GHS | GHS | GHS | GHS |
| Sales | 13,187.25 | 9559.919 | 2000 | 13,200 | 31,915.5 |
| Less Cost of Sales | 1615.705 | 738.5952 | 0 | 0 | 125 |
| Gross profit | 11,571.54 | 8821.324 | 2000 | 13,200 | 31,790.5 |
| Less Expenses | | | | | |
| Transportation | 937.3033 | 332.3333 | 0 | 0 | 100 |
| Weedicides | 1689.293 | 1466.083 | 0 | 0 | 250 |
| Labour | 1496.912 | 789.2857 | 150 | 0 | 250 |
| Depreciation | 158.8987 | 726.3333 | 80 | 80 | 225 |
| Others | 801.5069 | 492.3929 | 0 | 0 | 100 |
| Net Profit | 6487.629 | 5442.122 | 1770 | 13,120 | 30,865.5 |
| Cash Inflow | 13,187.25 | 9559.919 | 2000 | 13,200 | 31,915.5 |
| Cash Outflow | 6540.72 | 3391.464 | 150 | 0 | 1050 |

However, sales of staple crops are slightly higher (GHS 3627) than those of cash crops (Table 5). This may explain why people invest in staple crops. Moreover, staple crops provide food for households. This also explains why the cost of sales in staple crop production is high compared to that in cash crop production. High sales in staple crops may occur because different crops are planted on the same land.

The sales of farmers in the taungya system (on-reserve) are slightly more than two-fifths than those with a contractual relationship with other forestry companies (Table 5). This may be because more than 60% of the revenues of timber trees on the reserve do not go to the farmers. This explains why woodlot producers in a contractual relationship with other forestry companies receive more than twice the profit of those on the taungya system.

A woodlot producer who has no contractual relationship with other forestry companies makes a profit of GHS 1770 (Table 5) on timber trees. Farmers in the taungya system and woodlot producers with no contractual relationship do not incur as many expenses compared to off-reserve woodlot producers who have a contractual relationship with other companies.

### 3.2. NPV and BCR of Ventures in Ghana

Both agriculture and woodlot production are profitable (Table 6). This is because the NPV of all ventures is more than zero and the BCR is more than one (Figure 2). Although the BCRs of staple crop and woodlot production without any contractual relationship are similar (1.74 and 1.94 respectively), the NPVs of these ventures are different.

**Table 6.** NPV and BCR of agriculture and woodlot production.

| Type of Venture | NPV | BCR |
|---|---|---|
| | **GHS** | |
| Staple crop production | 45,716.64 | 1.94 |
| Cash crop | 40,944.03 | 2.49 |
| Woodlot without any contractual relationship | 17,387.11 | 1.74 |
| Woodlot under the taungya system | 41,486.75 | 14.11 |
| Woodlot with contractual relationship off-reserve | 628,739.39 | 6.76 |
| Woodlot farmers who are in associations | 2154.57 | 1.18 |

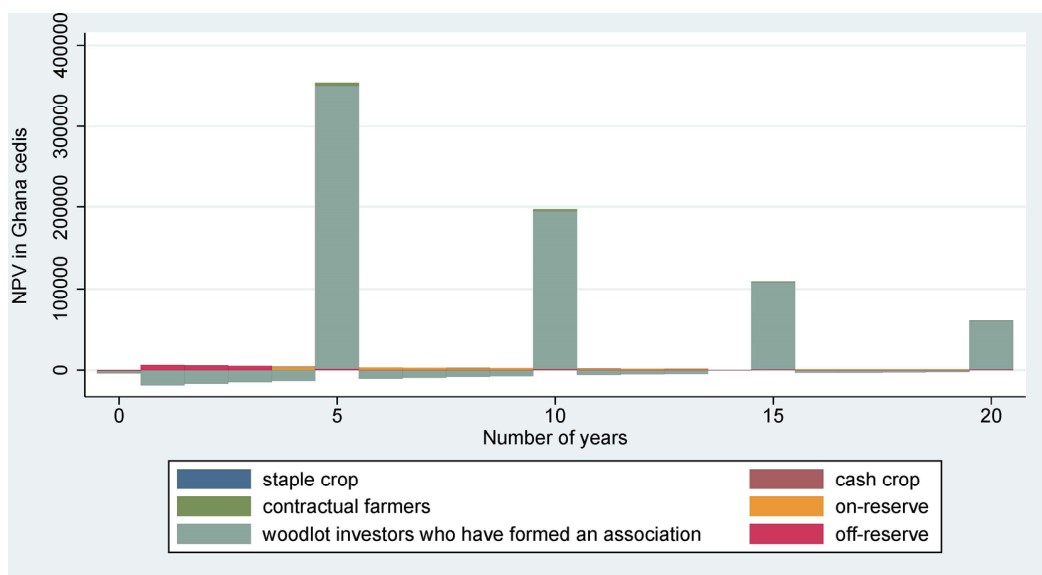

**Figure 2.** NPV of various ventures.

The NPV of the staple crop is GHS 49,590.63 and the BCR is 1.73 (Table A4).

The venture that provides the highest profit is the off-reserve woodlot with a contractual relationship (628,739.9) followed by staple crop production (Table 6). This shows that it is profitable for one to undertake staple crop production. The venture with the lowest NPV (2154.57) and BCR (1.18) is that of woodlot farmers who are in associations. This is because they pool their resources and thus increase cash outflow.

Cash crops farmer obtain an NPV of 40,944.03 and a Benefit-Cost Ratio of 2.49 (Table A5). This shows that it is more profitable for farmers to be involved in staple crops compared to cash crop production. Investors become involved in cash crop production because the price is stabilized and the government provides subsidies.

The NPV of investing in a woodlot without any contractual agreement is 17,387.11 and the BCR is 1.74 (Table A6). On-reserve farmers obtain an NPV of 41,486.75 and a BCR of 14.11 (Table A7), making this the most lucrative venture among all of the types of venture in terms of BCR after those on contractual relations (Table A8). The high BCR may be because the cash outflow of taungya farmers is low as a result of government subsidies. Investors who are in association generated the lowest NPV (Table A9).

## 4. Discussion

Cocoa production (in Ashanti and Western Regions) earns more revenue (10,960) than cashew (in the Bono-East region) (Table 3). However, expenses for cocoa are 3361.477 more than those of cashew (Table 3). This confirms the production theory that, as variable cost increases, total output increases [27]. Many cocoa farmers in the Bono-East Region are moving from the production of cocoa to cashew because of the amount of money spent to control pests and diseases on cocoa farms. Although the sales of cashew are low compared to those of cocoa, the low cash outflow in cashew makes it an attractive venture for investors.

Staple crop production is more profitable than cash crop production. This confirms [28] findings that farmers are more food and income secure when they are involved in cash crop production. During the first three years, when there is no revenue from cash crops, staple crops are planted so farmers can receive revenue during this period. With the development of a new variety of cash crops, farmers can harvest their cash crops and earn revenue from these crops as early as three years after production. Farmers enter into cash crops because there is a fixed price set by the government and subsidies are provided. This implies that most investors would use forest land for the cultivation of staple crops since profit in staple crop production is high. However, investors use forestlands for cash crops because of the price stability and subsidy provided by the government. This result confirms [9] findings that show land is put to the use that maximizes the present value of profits to the decision maker. This may explain why investors invest in staple crops and cash crops, but not woodlot production, thus leading to deforestation [2,3]. It may also explain why forestland is used for agriculture in the country [6,10].

Woodlot investors in associations have the lowest NPV and BCR values. This is because they contribute each month and ensure that every member is at par. Funds are generated internally to run the administrative cost of the association. These investors do not enjoy a fixed price or government subsidies, resulting in a low NPV. Investors without any contractual agreement have the lowest NPV and BCR values, except for those in associations. This confirms [5] findings that the forest does not generate sustainable revenue for investors. Their sales are low because of the low-quality timber logs they produce. They usually burn their lands each year. This is because they do not receive any incentive to pay labor costs or to buy weedicides to spray on the land. Soil [29] and wood quality are lost through the burning of forest lands each year; thus, people buy timber logs for as low as GHS 10. The comparison of woodlot production without any contractual agreement, with staple and cash crops, showed that staple and cash crops are more profitable than woodlot production. This is because farmers receive GHS 28,329.53 and 23,557.22 more in terms of NPV in staple and cash crops, respectively. This explains why farmers cut down timber trees on their cocoa farms [30]. Trees compete with staple and cash crops for space and nutrients, but provide less revenue to the farmer. This highlights the difficulties in agroforestry in Ghana.

Woodlot investors in contractual relationships (with companies or the government) have higher NPVs than those who are not in any contractual relationship. This confirms the finding that contractual relationships are good for agriculture [31] and the forestry industry [13]. When there is a contractual relationship between either the government or forestry companies, it becomes more profitable for one to venture into woodlot production. The cash outflow of investors in a contractual relationship with other forestry companies is high compared to the cash outflow of those who are not on contract and the taungya. This may be because the forestry companies who buy woodlots pay some expenses for investors. In previous studies [25,32], higher expenses were incurred in the production of the woodlot. This shows that there is a high investment when one produces woodlot on a contractual basis. Under the taungya system, the government absorbs most of the expenses. The cash outflow of investors in the taungya system is similar to the cash outflow of investors who are not in a contractual relationship with other firms. The cash outflow in crop production is 14 times higher than the cash outflow in woodlot production. This may be because the expenses in woodlot production considered in this study are highly absorbed and, where possible, minimized. A producer in a contractual relationship receives 16 times more revenue than one who is not in a contractual relationship with other companies. This means that, to earn more revenue in woodlot production, one has to invest significantly in its expenses.

The NPV of woodlot producers without any contractual relationship is low compared to that in previous studies [25,32]. This may be because the market is determined by demand and supply in Ghana and farmers do not know the value of timber logs. When one ventures into woodlot production, it takes about 20 years for trees to mature. An investor does not see the incentive to invest in woodlot production without any contractual relationship when it takes a very long time for trees to mature and there is a better alternative (agriculture) for one to invest in. About 60% of woodlot producers in the Bono-East Region have used their land for cashew production.

## 5. Conclusions

It is more profitable to venture into crop production than woodlot production when there is no contractual relationship. However, woodlot producers in contractual relationships with the Forest Commission and other forestry companies produce the highest Net Present Value (NPV) and Benefit-Cost Ratio (BCR). Nonetheless, this profitability is marginally higher than that of agriculture, which gives a fixed yearly return. This means woodlot production may not be a panacea to reducing agriculture in Ghana.

The existing tree-planting programs, such as the Green Ghana Project and Agroforestry, should investigate forming contractual relationships with investors to ensure sustainability.

**Author Contributions:** J.N.; Data curation, J.N.; Formal analysis, J.N.; Funding acquisition, J.N.; Investigation, J.N.; Methodology, J.N.; Software, J.N.; Visualization, J.N.; Writing—original draft, I.S.E.; Project administration, I.S.E.; Resources, I.S.E., A.M.-B. and E.E.O.; Supervision, I.S.E.; Validation, I.S.E.; Writing—review and editing. All authors have read and agreed to the published version of the manuscript.

**Funding:** This research was funded by Social and Environmental Trade-offs in African Agriculture (SENTINEL) of the Global Challenges Research Fund (GCRF)-UK through the Regional Universities Forum for Capacity Building in Agriculture (RUFORUM).

**Institutional Review Board Statement:** The study was conducted in accordance with the Declaration of Helsinki, and approved by the Ethics Committee of University of Ghana ECBAS 04/20.2021.

**Data Availability Statement:** Not applicable.

**Acknowledgments:** We are grateful to Anthony Egeru, David Ekepu, Adams Devenish and Robert Asiimwe for their support.

**Conflicts of Interest:** The authors declare no conflict of interest.

## Appendix A

**Table A1.** Profit and loss of various staple and cash crops in the Western region of Ghana.

| | Staple | | | | Cash Crop |
|---|---|---|---|---|---|
| | Plantain | Cassava | Others | Totals | Cocoa |
| | GHS | GHS | GHS | GHS | GHS |
| Sales | 4908 | 820 | 702 | 6430 | 5610 |
| Less Cost of Sales | 400 | 850 | 170 | 1420 | 400 |
| | 4508 | −30 | 532 | 5010 | 5210 |
| Less Expenses | | | | 0 | |
| Transportation | 100 | 200 | 30 | 330 | 280 |
| Weedicides | | 600 | 300 | 900 | 1000 |
| | | | | 0 | |
| Labor | 130 | 450 | 96 | 676 | 900 |
| Depreciation | 60 | 60 | 39 | 159 | 289 |
| Others | | 170 | 160 | 330 | |

**Table A2.** Profit and loss of various staple and cash crops in the Bono-East Region of Ghana.

| | Yam | Maize | Millet | Others | Totals | Cash Crop |
|---|---|---|---|---|---|---|
| | GHS | GHS | GHS | GHS | GHS | GHS |
| Sales | 7596.53846 | 2785.45455 | 1759 | 2640 | 14,780.993 | 3569.75758 |
| Less Cost of Sales | 495.238095 | 766.176471 | 107 | 237.25 | 1605.66457 | 215.785714 |
| | | | | | 0 | |
| Less Expenses | | | | | 0 | |
| Transportation | 307.954545 | 183.5 | 380 | 291 | 1162.45455 | 157 |
| Weedicides | 0 | 687.541667 | 705 | 492.272727 | 1884.81439 | 398.25 |
| Labor | 727.368421 | 323.333333 | 246.666667 | 270.714286 | 1568.08271 | 678.571429 |
| Depreciation | 68.0777778 | 45.3851852 | 22.6925926 | 22.6925926 | 158.848148 | 68.0777778 |
| Others | 462.117647 | 180 | 50 | 237.25 | 929.367647 | 215.785714 |
| Net Profit | 5535.78197 | 599.51789 | 247.640741 | 1088.82039 | 7471.761 | 1836.28694 |
| Cash Inflow | 7596.53846 | 2785.45455 | 1759 | 2640 | 14,780.993 | 3569.75758 |
| Cash Outflow | 1992.67871 | 2140.55147 | 1488.66667 | 1528.48701 | 7150.38386 | 1665.39286 |

**Table A3.** Profit and loss of various staple and cash crops in the Ashanti region of Ghana.

| | Maize | Cassava | Others | Totals | Cocoa |
|---|---|---|---|---|---|
| | GHS | GHS | GHS | GHS | GHS |
| Sales | 3000 | 2484 | 1900 | 7384 | 19,500 |
| Less Cost of Sales | 900 | 100 | 500 | 1500 | 1600 |
| Gross profit | 2100 | 2384 | 1400 | 5884 | 17,900 |
| Less Expenses | | | | | |
| Transportation | 190 | 380 | 200 | 770 | 560 |
| Weedicides | 120 | 245 | 110 | 475 | 3000 |
| Labor | 600 | 400 | 140 | 1140 | |
| Depreciation | 100 | 50 | 40 | 190 | 1890 |
| Others | 190 | 60 | 20 | 270 | 769 |
| Net Profit | 900 | 1249 | 890 | 3039 | 11,681 |
| Cash Inflow | 3000 | 2484 | 1900 | 7384 | 19,500 |
| Cash Outflow | 2000 | 1185 | 970 | 4155 | 5929 |

**Table A4.** NPV and BCR of farmers engaged in staple crop production.

| | Staple Crops | | | | | | |
|---|---|---|---|---|---|---|---|
| Number No. | Calendar Year | Cash Outflow | Cash Inflow | | Discounted Cash Outflow | Discounted Cash Inflow | |
| | | GH | GH | | GH | GH | GHS |
| 0 | 2020 | 376.50 | 0.00 | 1.00 | 376.50 | 0.00 | −376.50 |
| 1 | 2021 | 6540.72 | 13,187.30 | 0.89 | 5821.24 | 11,736.70 | 5915.46 |
| 2 | 2022 | 6541.72 | 13,188.30 | 0.80 | 5233.38 | 10,550.64 | 5317.26 |
| 3 | 2023 | 6542.72 | 13,189.30 | 0.71 | 4645.33 | 9364.40 | 4719.07 |
| 4 | 2024 | 6543.72 | 13,190.30 | 0.64 | 4187.98 | 8441.79 | 4253.81 |
| 5 | 2025 | 6544.72 | 13,191.30 | 0.57 | 3730.49 | 7519.04 | 3788.55 |
| 6 | 2026 | 6675.61 | 12,927.47 | 0.51 | 3404.56 | 6593.01 | 3188.45 |
| 7 | 2027 | 6675.61 | 12,927.47 | 0.45 | 3004.03 | 5817.36 | 2813.34 |
| 8 | 2028 | 6675.61 | 12,927.47 | 0.40 | 2670.25 | 5170.99 | 2500.74 |
| 9 | 2029 | 6675.61 | 12,927.47 | 0.36 | 2403.22 | 4653.89 | 2250.67 |
| 10 | 2030 | 6675.61 | 12,927.47 | 0.32 | 2136.20 | 4136.79 | 2000.60 |
| 11 | 2031 | 6806.51 | 12,663.65 | 0.29 | 1973.89 | 3672.46 | 1698.57 |
| 12 | 2032 | 6806.51 | 12,663.65 | 0.26 | 1769.69 | 3292.55 | 1522.86 |
| 13 | 2033 | 6806.51 | 12,663.65 | 0.23 | 1565.50 | 2912.64 | 1347.14 |
| 14 | 2034 | 6806.51 | 12,663.65 | 0.02 | 136.13 | 253.27 | 117.14 |
| 15 | 2035 | 6806.51 | 12,663.65 | 0.18 | 1225.17 | 2279.46 | 1054.29 |
| 16 | 2036 | 6937.40 | 12,399.82 | 0.16 | 1109.98 | 1983.97 | 873.99 |
| 17 | 2037 | 6937.40 | 12,399.82 | 0.15 | 1040.61 | 1859.97 | 819.36 |
| 18 | 2038 | 6937.40 | 12,399.82 | 0.13 | 901.86 | 1611.98 | 710.11 |
| 19 | 2039 | 6937.40 | 12,399.82 | 0.12 | 832.49 | 1487.98 | 655.49 |
| 20 | 2040 | 6937.40 | 12,399.82 | 0.10 | 693.74 | 1239.98 | 546.24 |
| | | | | | 48,862.24 | 94,578.88 | 45,716.64 |
| | | NPV | 45716.64 | | BCR | 1.94 | |

**Table A5.** Cash flow of farmers involved in cash crops.

| Number No. | Calendar Year | Cash Outflow | Cash Inflow | | Discounted Cost | Discounted Benefit | Discounted Cash Flow |
|---|---|---|---|---|---|---|---|
| | | GHS | GHS | | GHS | GHS | GHS |
| 0 | 2020 | 2432.90 | 0.00 | 1.00 | 2432.90 | 0.00 | −2432.90 |
| 1 | 2021 | 3391.46 | 9559.92 | 0.89 | 3028.57 | 8537.01 | 5508.43 |
| 2 | 2022 | 3392.46 | 9559.92 | 0.80 | 2703.79 | 7619.26 | 4915.47 |
| 3 | 2023 | 3393.46 | 9559.92 | 0.71 | 2416.14 | 6806.66 | 4390.52 |
| 4 | 2024 | 3394.46 | 9559.92 | 0.64 | 2158.88 | 6080.11 | 3921.23 |
| 5 | 2025 | 3395.46 | 9559.92 | 0.57 | 1925.23 | 5420.47 | 3495.25 |
| 6 | 2026 | 3463.37 | 9368.72 | 0.51 | 1755.93 | 4749.94 | 2994.01 |
| 7 | 2027 | 3463.37 | 9368.72 | 0.45 | 1565.44 | 4234.66 | 2669.22 |
| 8 | 2028 | 3463.37 | 9368.72 | 0.40 | 1399.20 | 3784.96 | 2385.76 |
| 9 | 2029 | 3463.37 | 9368.72 | 0.36 | 1250.28 | 3382.11 | 2131.83 |
| 10 | 2030 | 3463.37 | 9368.72 | 0.32 | 1115.20 | 3016.73 | 1901.52 |
| 11 | 2031 | 3531.28 | 9177.52 | 0.29 | 1013.48 | 2633.95 | 1620.47 |
| 12 | 2032 | 3531.28 | 9177.52 | 0.26 | 907.54 | 2358.62 | 1451.08 |
| 13 | 2033 | 3531.28 | 9177.52 | 0.23 | 808.66 | 2101.65 | 1292.99 |
| 14 | 2034 | 3531.28 | 9177.52 | 0.02 | 72.39 | 188.14 | 115.75 |
| 15 | 2035 | 3531.28 | 9177.52 | 0.18 | 646.22 | 1679.49 | 1033.26 |
| 16 | 2036 | 3599.19 | 8986.32 | 0.16 | 586.67 | 1464.77 | 878.10 |
| 17 | 2037 | 3599.19 | 8986.32 | 0.15 | 525.48 | 1312.00 | 786.52 |
| 18 | 2038 | 3599.19 | 8986.32 | 0.13 | 467.89 | 1168.22 | 700.33 |
| 19 | 2039 | 3599.19 | 8986.32 | 0.12 | 417.51 | 1042.41 | 624.91 |
| 20 | 2040 | 3599.19 | 8986.32 | 0.10 | 374.32 | 934.58 | 560.26 |
| | | | | | 27,571.72 | 68,515.76 | 40,944.03 |
| | | NPV | 40944.03 | | BCR | 2.49 | |

**Table A6.** Profitability of woodlot farmers (off-reserve).

| Woodlot Production—Off-Reserve | | | | | | | |
|---|---|---|---|---|---|---|---|
| Project | Actual | With the Project | With the Project | Discount Factor for Project Year | | | |
| Number No. | Calendar Year | Cash Outflow | Cash Inflow | | Discounted Cost | Discounted Benefit | |
| | | GHS | GHS | | GHS | GHS | GHS |
| 0 | 2020 | 941.25 | 0.00 | 1.00 | 941.25 | 0.00 | −941.25 |
| 1 | 2021 | 9140.00 | 16,075.00 | 0.89 | 8162.02 | 14,354.98 | 6192.96 |
| 2 | 2022 | 9140.00 | 16,075.00 | 0.80 | 7284.58 | 12,811.78 | 5527.20 |
| 3 | 2023 | 9140.00 | 16,075.00 | 0.71 | 6507.68 | 11,445.40 | 4937.72 |
| 4 | 2024 | 0.00 | 0.00 | 0.64 | 0.00 | 0.00 | 0.00 |
| 5 | 2025 | 150.00 | 2000.00 | 0.57 | 85.05 | 1134.00 | 1048.95 |
| 6 | 2026 | 150.00 | 0.00 | 0.51 | 76.05 | 0.00 | −76.05 |
| 7 | 2027 | 150.00 | 0.00 | 0.45 | 67.80 | 0.00 | −67.80 |
| 8 | 2028 | 150.00 | 0.00 | 0.40 | 60.60 | 0.00 | −60.60 |
| 9 | 2029 | 150.00 | 0.00 | 0.36 | 54.15 | 0.00 | −54.15 |
| 10 | 2030 | 153.00 | 1960.00 | 0.32 | 49.27 | 631.12 | 581.85 |
| 11 | 2031 | 153.00 | 0.00 | 0.29 | 43.91 | 0.00 | −43.91 |
| 12 | 2032 | 153.00 | 0.00 | 0.26 | 39.32 | 0.00 | −39.32 |
| 13 | 2033 | 153.00 | 0.00 | 0.23 | 35.04 | 0.00 | −35.04 |
| 14 | 2034 | 153.00 | 0.00 | 0.02 | 3.14 | 0.00 | −3.14 |
| 15 | 2035 | 156.00 | 1920.00 | 0.18 | 28.55 | 351.36 | 322.81 |
| 16 | 2036 | 156.00 | 0.00 | 0.16 | 25.43 | 0.00 | −25.43 |
| 17 | 2037 | 156.00 | 0.00 | 0.15 | 22.78 | 0.00 | −22.78 |
| 18 | 2038 | 156.00 | 0.00 | 0.13 | 20.28 | 0.00 | −20.28 |
| 19 | 2039 | 156.00 | 0.00 | 0.12 | 18.10 | 0.00 | −18.10 |
| 20 | 2040 | 156.00 | 1920.00 | 0.10 | 16.22 | 199.68 | 183.46 |
| | | | | | 23,541.20 | 40,928.31 | 17,387.11 |
| | | NPV | 17,387.11 | | BCR | 1.74 | |

**Table A7.** Profitability of woodlot farmers (on reserve).

| Woodlot Production On-Reserve | | | | | | | |
|---|---|---|---|---|---|---|---|
| Number No. | Calendar Year | Cash Outflow | Cash Inflow | | Discounted Cash Outflow | Discounted Cash Inflow | |
| | | GHS | GHS | | GHS | GHS | GHS |
| 0 | 2020 | 376.50 | 0.00 | 1.00 | 376.50 | 0.00 | −376.50 |
| 1 | 2021 | 1650.00 | 10,175.00 | 1.00 | 1473.45 | 9086.28 | 7612.83 |
| 2 | 2022 | 1650.00 | 10,175.00 | 1.00 | 1315.05 | 8109.48 | 6794.43 |
| 3 | 2023 | 0.00 | 0.00 | 0.89 | 0.00 | 0.00 | 0.00 |
| 4 | 2024 | 0.00 | 0.00 | 0.80 | 0.00 | 0.00 | 0.00 |
| 5 | 2025 | 0.00 | 0.00 | 0.71 | 0.00 | 0.00 | 0.00 |
| 6 | 2026 | 0.00 | 0.00 | 0.64 | 0.00 | 0.00 | 0.00 |
| 7 | 2027 | 0.00 | 0.00 | 0.57 | 0.00 | 0.00 | 0.00 |
| 8 | 2028 | 0.00 | 0.00 | 0.51 | 0.00 | 0.00 | 0.00 |
| 9 | 2029 | 0.00 | 0.00 | 0.45 | 0.00 | 0.00 | 0.00 |
| 10 | 2030 | 0.00 | 0.00 | 0.40 | 0.00 | 0.00 | 0.00 |
| 11 | 2031 | 0.00 | 0.00 | 0.36 | 0.00 | 0.00 | 0.00 |
| 12 | 2032 | 0.00 | 0.00 | 0.32 | 0.00 | 0.00 | 0.00 |
| 13 | 2033 | 0.00 | 0.00 | 0.29 | 0.00 | 0.00 | 0.00 |
| 14 | 2034 | 0.00 | 0.00 | 0.26 | 0.00 | 0.00 | 0.00 |
| 15 | 2035 | 0.00 | 0.00 | 0.23 | 0.00 | 0.00 | 0.00 |
| 16 | 2036 | 0.00 | 0.00 | 0.02 | 0.00 | 0.00 | 0.00 |
| 17 | 2037 | 0.00 | 0.00 | 0.18 | 0.00 | 0.00 | 0.00 |
| 18 | 2038 | 0.00 | 0.00 | 0.16 | 0.00 | 0.00 | 0.00 |

**Table A7.** *Cont.*

| | Woodlot Production On-Reserve | | | | | | |
|---|---|---|---|---|---|---|---|
| Number No. | Calendar Year | Cash Outflow | Cash Inflow | | Discounted Cash Outflow | Discounted Cash Inflow | |
| | | GHS | GHS | | GHS | GHS | GHS |
| 19 | 2039 | 0.00 | 0.00 | 0.15 | 0.00 | 0.00 | 0.00 |
| 20 | 2040 | 0.00 | 264,000.00 | 0.13 | 0.00 | 27,456.00 | 27,456.00 |
| | | | | 0.12 | 3165.00 | 44,651.75 | 41,486.75 |
| | | NPV | 41,486.75 | 0.10 | BCR | 14.11 | |

**Table A8.** Profitability of woodlot farmers (farmers in contractual relationship with other forestry companies).

| | Woodlot Companies | | | | | | |
|---|---|---|---|---|---|---|---|
| Project | Actual | With the Project | With the Project | Discount Factor for Project Year | | | |
| Number No. | Calendar Year | Total Cost | Total Benefits | | Discounted Cost | Discounted Benefit | |
| | | GHS | GHS | | GHS | GHS | GHS |
| 0 | 2020 | 4000.00 | 0.00 | 1.00 | 4000.00 | 0.00 | −4000.00 |
| 1 | 2021 | 14,197.50 | 0.00 | 0.89 | 12,678.37 | 0.00 | −12,678.37 |
| 2 | 2022 | 14,197.50 | 0.00 | 0.80 | 11,315.41 | 0.00 | −11,315.41 |
| 3 | 2023 | 14,197.50 | 0.00 | 0.71 | 10,108.62 | 0.00 | −10,108.62 |
| 4 | 2024 | 14,197.50 | 0.00 | 0.64 | 9029.61 | 0.00 | −9029.61 |
| 5 | 2025 | 14,197.50 | 638,310.00 | 0.57 | 8049.98 | 361,921.77 | 353,871.79 |
| 6 | 2026 | 14,481.45 | 0.00 | 0.51 | 7342.10 | 0.00 | −7342.10 |
| 7 | 2027 | 14,481.45 | 0.00 | 0.45 | 6545.62 | 0.00 | −6545.62 |
| 8 | 2028 | 14,481.45 | 0.00 | 0.40 | 5850.51 | 0.00 | −5850.51 |
| 9 | 2029 | 14,481.45 | 0.00 | 0.36 | 5227.80 | 0.00 | −5227.80 |
| 10 | 2030 | 14,481.45 | 625,543.80 | 0.32 | 4663.03 | 201,425.10 | 196,762.08 |
| 11 | 2031 | 14,765.40 | 0.00 | 0.29 | 4237.67 | 0.00 | −4237.67 |
| 12 | 2032 | 14,765.40 | 0.00 | 0.26 | 3794.71 | 0.00 | −3794.71 |
| 13 | 2033 | 14,765.40 | 0.00 | 0.23 | 3381.28 | 0.00 | −3381.28 |
| 14 | 2034 | 14,765.40 | 0.00 | 0.02 | 302.69 | 0.00 | −302.69 |
| 15 | 2035 | 14,765.40 | 612,777.60 | 0.18 | 2702.07 | 112,138.30 | 109,436.23 |
| 16 | 2036 | 15,049.35 | 0.00 | 0.16 | 2453.04 | 0.00 | −2453.04 |
| 17 | 2037 | 15,049.35 | 0.00 | 0.15 | 2197.21 | 0.00 | −2197.21 |
| 18 | 2038 | 15,049.35 | 0.00 | 0.13 | 1956.42 | 0.00 | −1956.42 |
| 19 | 2039 | 15,049.35 | 0.00 | 0.12 | 1745.72 | 0.00 | −1745.72 |
| 20 | 2040 | 15,049.35 | 600,011.40 | 0.10 | 1565.13 | 62,401.19 | 60,836.05 |
| | | | | | 109,146.97 | 737,886.36 | 628,739.39 |
| | | NPV | 628,739.39 | | BCR | 6.76 | |

**Table A9.** Profitability of woodlot farmers (those who are in associations).

| | Woodlot Companies | | | | | | |
|---|---|---|---|---|---|---|---|
| Project | Actual | With the Project | With the Project | Discount Factor for Project Year | | | |
| Number No. | Calendar Year | Total Cost | Total Benefits | | Discounted Cost | Discounted Benefit | |
| | | GHS | GHS | | GHS | GHS | GHS |
| 0 | 2020 | 1349.00 | 0.00 | 1.00 | 1349.00 | 0.00 | −1349.00 |
| 1 | 2021 | 1400.00 | 0.00 | 0.89 | 1250.20 | 0.00 | −1250.20 |
| 2 | 2022 | 1400.00 | 0.00 | 0.80 | 1115.80 | 0.00 | −1115.80 |
| 3 | 2023 | 1400.00 | 0.00 | 0.71 | 996.80 | 0.00 | −996.80 |

<div align="center"><b>Table A9.</b> <i>Cont.</i></div>

| Woodlot Companies | | | | | | | |
|---|---|---|---|---|---|---|---|
| Project | Actual | With the Project | With the Project | Discount Factor for Project Year | | | |
| Number No. | Calendar Year | Total Cost | Total Benefits | | Discounted Cost | Discounted Benefit | |
| | | GHS | GHS | | GHS | GHS | GHS |
| 4 | 2024 | 1400.00 | 0.00 | 0.64 | 890.40 | 0.00 | −890.40 |
| 5 | 2025 | 1400.00 | 12,000.00 | 0.57 | 793.80 | 6804.00 | 6010.20 |
| 6 | 2026 | 1428.00 | 0.00 | 0.51 | 724.00 | 0.00 | −724.00 |
| 7 | 2027 | 1428.00 | 0.00 | 0.45 | 645.46 | 0.00 | −645.46 |
| 8 | 2028 | 1428.00 | 0.00 | 0.40 | 576.91 | 0.00 | −576.91 |
| 9 | 2029 | 1428.00 | 0.00 | 0.36 | 515.51 | 0.00 | −515.51 |
| 10 | 2030 | 1428.00 | 11,760.00 | 0.32 | 459.82 | 3786.72 | 3326.90 |
| 11 | 2031 | 1456.00 | 0.00 | 0.29 | 417.87 | 0.00 | −417.87 |
| 12 | 2032 | 1456.00 | 0.00 | 0.26 | 374.19 | 0.00 | −374.19 |
| 13 | 2033 | 1456.00 | 0.00 | 0.23 | 333.42 | 0.00 | −333.42 |
| 14 | 2034 | 1456.00 | 0.00 | 0.02 | 29.85 | 0.00 | −29.85 |
| 15 | 2035 | 1456.00 | 11,520.00 | 0.18 | 266.45 | 2108.16 | 1841.71 |
| 16 | 2036 | 1484.00 | 0.00 | 0.16 | 241.89 | 0.00 | −241.89 |
| 17 | 2037 | 1484.00 | 0.00 | 0.15 | 216.66 | 0.00 | −216.66 |
| 18 | 2038 | 1484.00 | 0.00 | 0.13 | 192.92 | 0.00 | −192.92 |
| 19 | 2039 | 1484.00 | 0.00 | 0.12 | 172.14 | 0.00 | −172.14 |
| 20 | 2040 | 1484.00 | 11,280.00 | 0.10 | 154.34 | 1173.12 | 1018.78 |
| | | | | | 11,717.43 | 13,872.00 | 2154.57 |
| | | NPV | 2154.57 | | BCR | 1.18 | |

## Note

[1] 1 Dollar = 7.5 Ghana Cedis. All monetary figures are in Ghana Cedis.

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
