# Peer review of "Financial Analysis of the Use of Land: Agriculture or Woodlot"

_land, doi:10.3390/land11050642_

Round 1

Reviewer 1 Report

The phrases are too short and the style is a little bit telegraphic. There are lots of short sentences that can be merged into a single meaningful phrase.

in addition to this overall observation, some suggestions:

line 97. Primary data was from questionnaire.  just one questionnaire?

lines 100-101: there must be some literature articles about deforestation, worth being quoted.

table 2: gender instead of sex.

line 121: young are not (employed) in farming production. This assumption is based on the surveys? what if the sampling was biased?

line 214: cashoutflow...

line 215: there is not predicate in the sentence! Cash outflow in crop production...

somewhere in the text the authors told something about the revenue from the woodlot, which is evenly distributed across the period. according to european forestry standards, there must be a sort of rotation, meaning that cash flows are not evenly distributed.

the reader has no clue about the exchange rate of the currency used in the article.

Author Response

The phrases are too short and the style is a little bit telegraphic. There are lots of short sentences that can be merged into a single meaningful phrase.

Addressed: some of the sentences have been merged

in addition to this overall observation, some suggestions:

line 97. Primary data was from the questionnaire.  just one questionnaire?.

Addressed: There were two sets of questionnaires: one for woodlot producers and the other for farmers. Each set had respondents 193

lines 100-101: there must be some literature articles about deforestation, worth being quoted.

Addressed: literature articles about deforestation have been quoted.

Table 2: gender instead of sex.

Addressed: sex has been removed

line 121: young are not (employed) in farming production. This assumption is based on the surveys? what if the sampling was biased?

line 214: cashoutflow...

Addressed: cashoutflow has been changed to cash outflow

line 215: there is no predicate in the sentence! Cash outflow in crop production...

Addressed: Cash outflow in crop production is 14 times higher than cash outflow in woodlot production.

somewhere in the text, the authors told something about the revenue from the woodlot, which is evenly distributed across the period. according to European forestry standards, there must be a sort of rotation, meaning that cash flows are not evenly distributed. Addressed: this has been Addressed

the reader has no clue about the exchange rate of the currency used in the article.

The exchange rate is at the footnote

Reviewer 2 Report

The subject is very interesting but the presentation doesn't.
 Figures and details from the study area should be improved. Once you divide the study area into 3 regions, you should present the results based on similar characteristics or about the regions. The text was very confusing to understand the differences between regions x producers (forest or crops).
The English language must be reviewed. There are words bad written and repetitive at the same phrase.
The data and analysis are good, but the presentation must be reviewed and better explained.

Author Response

The subject is very interesting but the presentation doesn't.
 Figures and details from the study area should be improved. Once you divide the study area into 3 regions, you should present the results based on similar characteristics or about the regions. The text was very confusing to understand the differences between regions x producers (forest or crops).
The English language must be reviewed. There are words bad written and repetitive in the same phrase.
The data and analysis are good, but the presentation must be reviewed and better explained.

Addressed: Figures and details from the study area have been improved. Regional results have been presented and results are based on similar characteristics of the regions.

Reviewer 3 Report

The article is interesting and provides a pertinent analysis of the study on land use, in several variants, in Ghana.
The introduction is quite brief, but interesting.
The bibliography is acceptable and meets the criteria that are required for an article to be published in the journal Land.
The authors present several estimation methods and show which are more efficient in terms of land use, in various situations.
Interesting discussions and conclusions are presented, but quite brief, especially at the conclusions.
Profitability of forestry and agriculture in Ghana is presented and estimated.
The two variants with the advantages and disadvantages of each of the methods are presented.
The article is an interesting study and I appreciate that it can be accepted for publication, subject to those presented in the reviews in the review form.

Author Response

Manuscript has been improved